# Investigating the influence of the physical environment on psychiatric nurses wellbeing and professional interactions: A convergent parallel mixed-method study protocol

Milica Vujovic[1]*, Maja Kevdzija[2], Friedrich Neuhauser[3], Matthäus Fellinger[3,4,5]

1 Department of Digital Architecture and Planning, Faculty of Architecture and Planning, Institute of Architectural Sciences, TU Wien, Vienna, Austria, 2 Department of Building Theory by Design, Faculty of Architecture and Planning, Institute of Architecture and Design, TU Wien, Vienna, Austria, 3 2nd Department of Psychiatry and Psychotherapy, Hietzing Clinic, Vienna Healthcare Group, Vienna, Austria, 4 Karl Landsteiner Institute for Mental Health, Vienna, Austria, 5 Clinical Division of Social Psychiatry, Medical University of Vienna, Vienna, Austria

☯ These authors contributed equally to this work.
* milica.vujovic@tuwien.ac.at

## Abstract

Nurses in psychiatric hospitals face demanding working conditions, where high emotional burdens, unpredictable patient behavior, and safety concerns contribute to stress. Research highlights the significant impact of the physical environment on healthcare workers' experiences, with environmental factors either reducing or increasing stress levels, affecting professional interactions, and influencing job satisfaction and well-being. Despite growing recognition of these effects, there remains a need for robust, evidence-based methodologies to systematically examine how specific spatial features impact psychiatric nurses' experiences. This study protocol aims to develop and test scalable and structured methods to evaluate the influence of the physical environment on stress levels, professional interactions, and spatial perceptions among psychiatric nurses. Conducting research in psychiatric hospitals presents unique challenges, requiring methodologies that consider the sensitive nature of the setting. Issues such as patient confidentiality, changeable work dynamics, and psychological demands on staff require a carefully tailored data collection approach. This study protocol addresses outlined concerns by selecting methods and equipment that minimize disruptions while ensuring reliable and meaningful data collection. Observations are conducted unobtrusively, and the study design prioritizes ethical considerations, ensuring participant comfort and patient privacy. The study employs a convergent parallel mixed methods design, simultaneously collecting and analyzing qualitative and quantitative data. Quantitative measures include wearable sensors tracking heart rate variability, light exposure, and physical activity alongside surveys and systematic in-person observations of professional interactions. Qualitative data, gathered through semi-structured interviews, explores nurses' perceptions of their

**Data availability statement:** No datasets were generated or analysed during the current study. All relevant data from this study will be made available upon study completion.

**Funding:** TU Wien Bibliothek provides financial support through its Open Access Funding Programme.

**Competing interests:** The authors have declared that no competing interests exist.

work environment and stress levels. The study aims to provide new insights relevant to interdisciplinary research, demonstrating the value of integrating architectural and healthcare perspectives to improve the design of therapeutic and work-friendly environments.

## Introduction

Psychiatric nursing is a critical component of mental health care, demanding a unique combination of clinical acumen, empathy, and alertness. These professionals navigate a diverse and intricate range of responsibilities, from crisis management and ongoing patient support to medication administration and therapy facilitation. This multifaceted role necessitates not only advanced clinical competencies but also significant emotional fortitude [1–7]. The nature of psychiatric care often exposes nurses to aggressive behaviours and emotionally demanding situations, contributing to high-stress levels [5,8–12]. These stressors, combined with heavy workloads and the constant need for patient attention, can negatively impact mental health, job satisfaction, and the quality of care provided [13,14]. As frontline caregivers, nurses often must balance patient care with resource management [15], making their working conditions particularly demanding.

In psychiatric settings, various factors influence stress levels and perceptions of the work environment. Evidence-based design and environmental psychology research consistently highlight the significant impact of physical environments on individuals and their activities in the healthcare context [16–18]. The impact of medical facility environments on staff, especially nurses, is gaining attention due to their frequent interaction with patients and surroundings [16–21]. However, the effect of the physical environment on psychiatric nurses is still underexplored. Understanding the link between stress, the environment, and interpersonal dynamics is key to enhancing nurse well-being and patient care.

This study aims to contribute to closing this gap by focusing on psychiatric nurses and presents a study protocol where a robust and scalable methodology is developed, which enables the application of research in a sensitive context, does not interfere with the work of nurses and covers different aspects of the physical environment.

### Workplace stress in psychiatric nursing

Psychiatric nurses face unique challenges in their workplace, including emotional, physical, and psychological demands, which differ from those in general medical settings [21,22]. In addition to patient care, their responsibilities often extend to administrative tasks, ward organisation, and decision-making, particularly in managerial roles [13,23]. Avoiding the over-generalisation of psychiatric nurses' roles is essential for developing research methodologies that accurately capture the impact of the environment and behaviour on their work.

The nurse-patient alliance is central to the therapeutic process [24,25]. However, managing multiple patients with complex conditions, alongside administrative

burdens and other stressors, can contribute to elevated stress and the risk of burnout [26]. Psychiatric nurses are also at a higher risk of trauma exposure, with increased susceptibility to PTSD [27,28]. Notably, Soravia et al. [27] reported significantly higher stress levels among psychiatric nurses and other rescue workers—such as police officers, firefighters, ambulance personnel, and emergency responders—who were confronted with patient suicide, compared to those who were not.

Given the proven impact of the physical environment on an individual's wellbeing, it is important to examine how the environment contributes to stress levels and overall work performance among psychiatric nurses. Understanding how the physical environment interacts with other stressors could lead to interventions that enhance the work environment and improve nurses' well-being.

### Nurse-patient interactions in psychiatric hospitals

Psychiatric nurses work with a diverse range of patients, including individuals with dementia, intellectual disabilities, psychiatric disorders, and those involved in forensic psychiatry care. Common behaviors like aggression, suicidal tendencies, and depression require a wide range of therapeutic skills [3,26,27,29]. Communication is key to these interactions, fostering therapeutic relationships, though negative patient responses can lead to frustration and complicate care [23,25,26].

Research on nurse-patient interactions primarily relies on qualitative data, often focusing on either nurses' or patients' perspectives through interviews [30–32]. Koivisto et al. [33] found that patients often felt cared for but perceived care as unstructured, leading to confusion about their treatment and staff availability. While some studies highlight inadequate interaction between patients and nurses, differences in context and available resources make it challenging to generalize these findings [23,34].

To our knowledge, the specific locations within psychiatric wards where nurse-patient interactions occur and the characteristics of these environments have not been sufficiently studied. Given the significant influence of the environment on individuals, it is essential to explore this relationship further. Additionally, examining how the environment and interactions relate to other factors, such as stress and perceptions of the ward, could provide valuable insights.

### Effects of the physical environment on healthcare workers

In their literature review, Gharaveis et al. [19] emphasised the physical environment's significant impact on teamwork and communication of healthcare staff, highlighting its indirect effect on care quality. Furthermore, the physical environment can influence patients' mental health recovery, which in turn impacts healthcare staff well-being and performance [35,36]. Alongside the social environment and healthcare delivery model, the physical environment is crucial, where spatial characteristics, including furnishings, seating arrangements, visibility, accessibility, private spaces, and workstation locations, are identified as key elements that enhance teamwork and communication [16–18].

Different studies have focused on different aspects of the physical environment. Visibility, a prerequisite for good communication between staff, is an essential parameter for designing healthcare spaces and contributes to reducing stress and walking distance [19–21,37]. It also provides the possibility to perform several tasks at the same time because it enables the monitoring of a larger area.

However, communication between staff differs significantly between departments due to the different nature of patients' conditions [19], and visibility, as well as other factors and their impact on nursing staff, have yet to be sufficiently investigated in the context of psychiatric hospitals.

When it comes to psychiatric hospitals and wards, and their effect on nursing staff, studies are scarce. Tyson et al [38] report on the increase in positive interactions between patients and nurses, in a studied newly redesigned ward, with indications of an increase in burnout. The environment that implies physical restrictions has been common on psychiatric wards, which indirectly contributes to nurses' stress as patients display a greater number of aggressive episodes and dissatisfaction [39].

Despite the critical role of psychiatric nurses, there is a lack of comprehensive studies addressing the relationship between their physical work environment and stress levels. Existing research often needs more depth and breadth to understand these dynamics fully.

## Methodological challenges in psychiatric nursing research

Research regarding the impact of the environment on individuals requires an understanding of both objective and subjective aspects [20]. A convergent parallel mixed-method design integrates quantitative and qualitative approaches, offering a comprehensive understanding of complex research questions [40–42]. This methodology addresses not only questions of quantity and frequency but also delves into the processes and reasons behind them, providing deeper insights in contexts where both numerical outcomes and lived experiences are crucial. By collecting data simultaneously, this approach enhances time efficiency. Additionally, the cross-validation of quantitative and qualitative findings strengthens the study's overall validity [40,41].

Research examining the influence of healthcare environments frequently employs a diverse array of methodological approaches. Rafeeq and Mustafa [20] integrated qualitative and quantitative methods, utilizing a checklist tool to evaluate design quality (qualitative method) alongside questionnaires and space syntax analysis (quantitative methods). Quantitative data collection focused on key environmental parameters, including visual and physical connectivity, privacy, accessibility, noise levels, positive distractions, and safety and security.

Understanding the interaction between healthcare staff and patients has also necessitated using qualitative and quantitative instruments. For instance, a quantitative tool measuring the working alliance was employed to investigate associations and discrepancies in the perspectives of nurses and patients concerning therapeutic relationships [25]. The study highlighted limitations in establishing causal relationships among variables, underscoring the inherent complexity of mental health research and the necessity of multimethod approaches. Similarly, research into nurses' perceptions of empathetic nurse-patient communication adopted qualitative methodologies, employing content analysis of interviews conducted with nurses [31]. In another example, Gharaveis et al. [19] combined observational data and interviews to assess how different types of visibility influence the efficiency of medical professionals in emergency department settings.

Investigating the effects of healthcare environments requires an interdisciplinary perspective, as the multifaceted nature of these research questions often demands the integration of diverse methodologies. Mixed-method approaches, as demonstrated in prior studies, are particularly well-suited for addressing this complexity. The sensitive and challenging context of healthcare further necessitates careful and deliberate selection of methods to ensure robust and meaningful findings.

## Research questions and expected outcomes

The protocol presented in this paper is designed for the study that aims to explain the relationship between the physical work environment and stress levels of nursing staff in psychiatric hospitals. The study also considers the complexity and diversity of nurses' tasks and their additional roles in administration and decision-making. Therefore, the presented protocol aims to offer a methodology that delves deeper into the nuances of their roles and provides data on the impact of the physical environment. The findings are expected to highlight the role of the physical environment in the well-being of healthcare workers and provide evidence that would further support the architectural design of healthcare environments, specifically psychiatric hospitals. Furthermore, the study protocol aims to provide a methodology that is reproducible and scalable. A mixed-method study employing a convergent parallel design aims to address the overarching research question "How does the physical environment of psychiatric hospitals influence the well-being and professional interactions of nursing staff? More specific research questions are further developed:

1. What is the interplay between the characteristics of the physical environment and the stress levels of nursing staff in psychiatric hospitals?

2. What is the interplay between the characteristics of the physical environment and the interactions of nursing staff with patients, family members, and other medical staff in psychiatric hospitals?

3. How do nurses working in psychiatric hospitals perceive their working environment in relation to their well-being?

## Method and analysis

### Study setting

The study is conducted in the 2nd Department of Psychiatry at Hietzing Clinic in Vienna, Austria, which consists of four different units and involves nursing staff, in the period between May 2024 and September 2025. All four units units are involved in the study: 1) a unit for acute psychiatric admissions for patients aged 18–49, 2) a unit for acute psychiatric admissions for patients aged 50 and above, 3) a transpsychiatric unit for patients aged 16–25 and 4) a unit for people with intellectual disability and psychiatric disorders. Each unit treats a range of patient conditions. All four units are organised in the same way and consist of patient rooms with en-suite bathrooms, doctors' offices, a medication room, patients' dining room, nurses' station, kitchen, nurses' break room, staff toilets, a risk room, a winter garden for patients, a meeting room, and storage rooms. Nursing staff participating in the study are recruited from all four units. Furthermore, participants may be recruited not only from the inpatient wards but also from the ambulance, day care, and garden therapy units, ensuring a comprehensive sample.

The study setting is selected based on the hospital's ability to provide both a sufficient number of participants and a rich dataset. In addition to its standard psychiatric units, the hospital also includes a garden therapy program with a unique space dedicated to this activity, allowing for various spatial features to be included in the analysis. By incorporating these diverse environmental characteristics, more meaningful insights can be gained, allowing for a thorough exploration of how different spatial settings relate to participant data and more effectively addressing the research questions. Researchers were introduced to the staff and the collaboration was established before data collection began, ensuring access of the researchers to device adjustment and proper data collection. Furthermore, prior to data collection, researchers were introduced to the units' floor plans and the staff–patient dynamics to ensure their presence caused minimal disruption to clinical activities.

### Study design

The proposed study employs a mixed-methods approach using a convergent parallel design [41]. It involves the simultaneous collection of qualitative and quantitative data, which will be analyzed separately and then integrated to provide a comprehensive and robust understanding of the research topic. Each dataset is analysed separately using methods suited to its format, i.e., statistical analysis for sensor and physiological data, and thematic coding for interview and observational material. Findings are then aligned through joint displays and comparison matrices to identify convergence, complementarity, or discrepancy, supporting an integrated interpretation grounded in both experiential and objective perspectives. This analytical structure enables environmental measures and physiological indicators to be directly compared with staff experiences and perceptions, ensuring that subjective reports are neither assumed nor dismissed, but rather examined alongside empirical observations. Participant and environmental data, including stress levels, sleep patterns, light exposure, environmental perceptions, and physical environmental features, are gathered using various measurement tools to evaluate study outcomes (Fig 1). Qualitative data, derived from observations and interviews, will capture participants' experiences and specific aspects of behaviour and workflow. At the same time, quantitative metrics such as heart rate variability (HRV) and light exposure measurements will provide objective insights into physiological responses and environmental factors. Integrating these methods will enable the identification of correlations between stress levels and environmental influences, revealing patterns that might not be apparent when using either approach in isolation. This study aims to address critical gaps in understanding how the physical environment impacts the well-being of psychiatric

Data

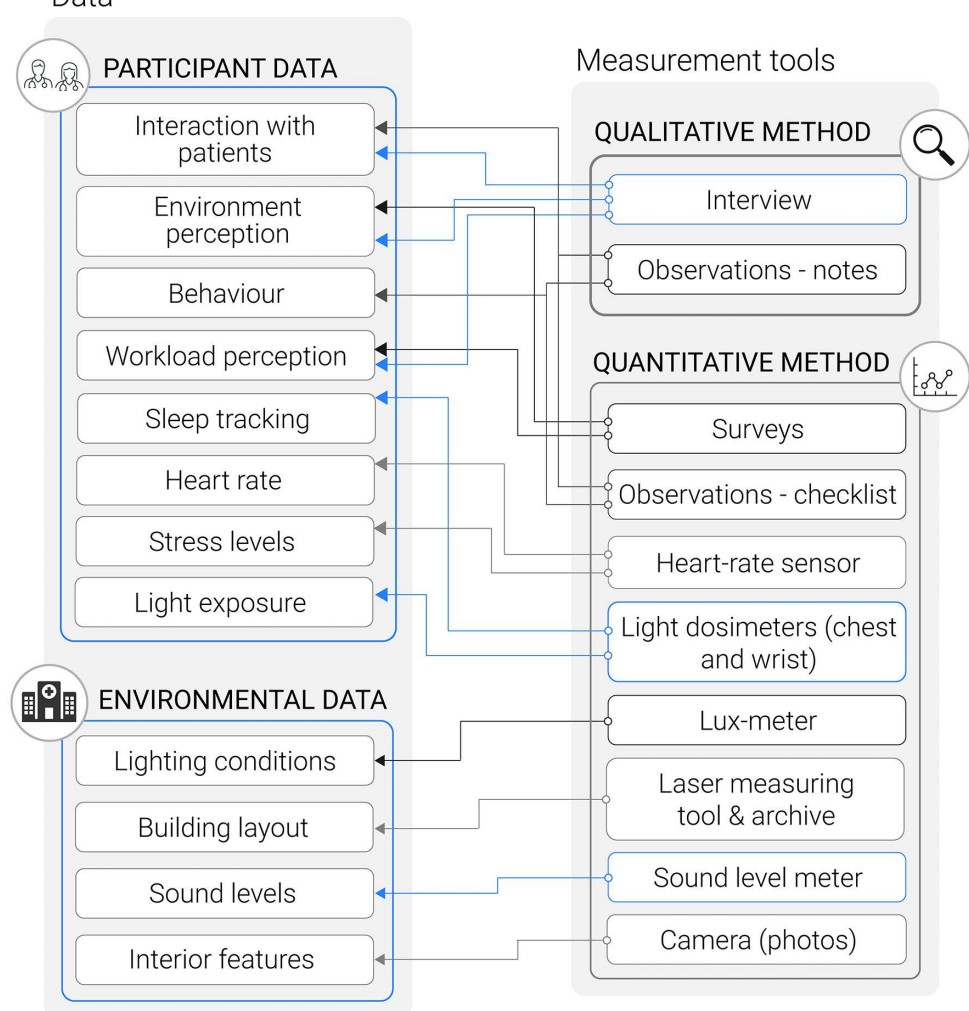

**Fig 1. Diagrammatic overview of the mixed-method approach in data collection with quantitative and qualitative methods applied.**

hospital staff, a unique and sensitive working population. Careful selection of non-invasive measurement tools is intended to ensure minimal disruption to the participants' workflow.

Since the study uses comprehensive methods including interviews, the data set for each individual participant will be rich in both objective and subjective data collected. The target sample size for this study is 20 participants. As the mixed-method approach focuses on an in-depth understanding of the effects of the environment on stress levels, the study employs a purposeful sampling that provides rich, detailed insights, even with smaller number of participants, which is considered sufficient for meaningful analysis [43–45]. Staff in a psychiatric hospital encounter various situations where the use of the methods explained below may present chalenges.

Since psychiatric hospitals are sensitive environments and certain limitations to research applicability may exist, this study protocol follows a methodologically reproducible framework that can be partially or fully replicated in subsequent studies. In this context, reproducibility refers to the capacity of other researchers to replicate the procedural design, measurement instruments, and analytical approach, rather than to reproduce the same participant sample or identical

datasets. While convenience and purposeful sampling inherently constrain statistical replication, the study ensures procedural transparency through detailed documentation of all protocols, sensor configurations, and analytic workflows. This methodological clarity enables the study methodology to be reapplied, tested, or adapted across comparable psychiatric or healthcare settings, thereby supporting consistency and comparability of results despite variations in sample composition. Therefore, this approach supports the scalability of the study protocol and its adaptation to different contexts and settings.

To ensure methodological transparency and reproducibility, the study follows clearly defined, stepwise procedures for data collection, processing, and analysis. All instruments, sensor configurations, and survey tools are standardized and documented to allow replication in other psychiatric or healthcare environments. The mixed-methods framework, with its parallel yet integrable data streams, supports consistent replication across settings with different spatial or organizational characteristics. Moreover, data handling protocols, ethical safeguards, and analytic procedures are designed to be adaptable, enabling other research teams to reproduce or extend the study while maintaining comparability of results. This systematic structure enhances scientific credibility and facilitates cumulative knowledge-building in environmental and mental health research.

Confounding variables in this study may include factors such as individual differences in resilience or prior trauma that could influence participants' stress levels and overall well-being. Through the integration of qualitative and quantitative data within a mixed-methods framework, the influence of confounding variables is reduced, enabling a more precise evaluation of how environmental factors relate to stress outcomes among psychiatric nurses. This reduction is achieved by triangulating multiple data sources, such as physiological measures, environmental sensor data, and participants' self-reported experiences, to cross-validate findings and distinguish environmental effects from individual differences.

## Study participants

Participants in this study are nurses employed and actively engaged in working activities within the 2nd Department for Psychiatry at Hietzing Clinic in Vienna, Austria. They are involved in patient care duties and coordination of work activities during the study period. Convenience sampling is applied and participation is voluntary with all details of the study made available to the potential participants. Recruitment began on May 1st 2024. Data collection is conducted on a per-subject basis, and the recruitment process is planned to continue until May 2025. Furthermore, participants are recruited through a contact person, who serves as an intermediary to avoid any bias that might be introduced if participants had direct contact with the hospital management involved in the study design. In this way, the hospital management is not directly involved in the recruitment process. Furthermore, participation in the study has no positive or negative impact on nurses' opportunities for career advancement within the workplace. Retention rates are expected to be high, as participants are employees at the psychiatric hospital. Due to the short duration of participation (minimum of two weeks) and detailed explanation of the study, it is expected that the participants who agree to take part will be dedicated to completing the study. The inclusion criteria are defined based on the research question, which aims to identify potential influences of the environment on stress levels and behaviour rather than connecting these influences with predetermined participant characteristics. Therefore, age, gender, or other characteristics are not considered exclusion criteria. As quantitative and qualitative data are collected simultaneously, all types of data are collected for the same number of participants.

The study takes place from May 2024 until September 2025 and involves a minimum of 7 shifts for each participant. Each nursing shift lasts between 8 and 12 hours, depending on the nurse's role within the ward hierarchy and their specific clinical responsibilities. To obtain a minimum of 20 hours of valid heart rate recordings and observational data during active shifts, and to capture reliable measures of sleep and light exposure over a continuous two-week period, an individualized data-collection schedule is developed for each participant. An example of the schedule for one participant is shown in Fig 2. This tailored scheduling approach ensures balanced coverage of different shift types (day, evening, and night). It

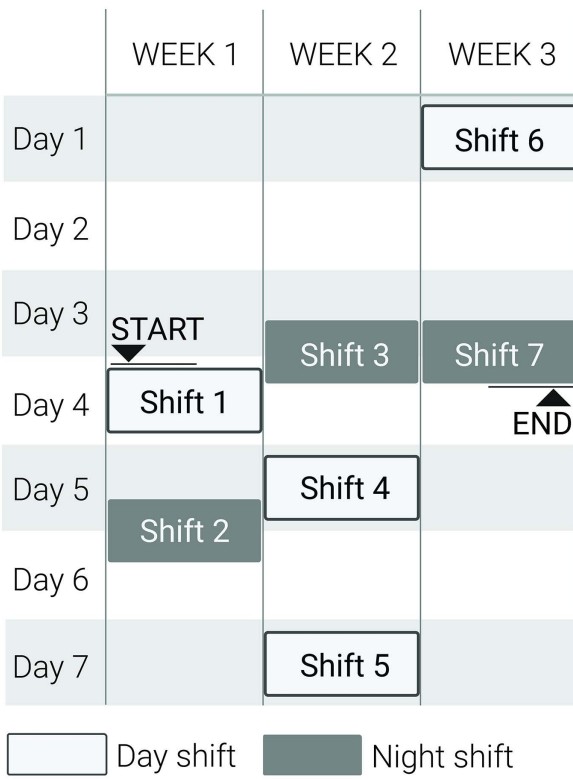

**Fig 2. Example of the schedule for one participant with 7 shifts (including day and night) across at least two weeks.**

accounts for variations in workload and rest periods, thereby improving data consistency and minimising missing or biased measurements. Nurses are selected to represent different levels of responsibility and duties, ensuring participation from various positions within the organizational framework. As the data collection is conducted sequentially for each participant, the participants are enrolled consecutively rather than simultaneously. At the same time, the study schedule is continually adjusted to align with each nurses' shift pattern. This method provides a rich database that enables detailed analysis and development of individual participant profiles. The study protocol is designed so that with additional equipment and resources, it is possible to carry out measurements with several participants simultaneously.

## Quantitative data

Researchers will begin by collecting baseline data, including demographic information, life habits, and job satisfaction through validated surveys. Using smart devices to establish individual baseline, they will also measure physiological parameters like heart rate (HR) and heart rate variability (HRV) in a relaxed state. Different research instruments are used in the study for data collection. Each method is explained in detail in the following sections, with the following overview:

• Heart rate measurements (for HRV calculation): Participants wear a heart rate sensor strap

• Light dosimetry: light dosimeters to track light exposure, sleep patterns. Participants wear them on the wrist and on the chest

• Environment characteristics: Researchers map the physical environment using various tools at specific times and conditions, including room dimensions, lighting, and sound levels.

- Systematic in-person observations: Observation checklist is used to collect real-time data on behaviours and interactions.

- Surveys: MEQ [46], CARE [47] and NASA Task Load Index [48,49], for assessing participants' well-being and work environment. Surveys are conducted in German.

- Semi-structured interviews: Interviews explore participants' experiences and perceptions of the physical environment, focusing on themes like spatial layout and lighting. Interviews are conducted in German.

   Duration of data collection varies based on the data type. Table 1 represents an overview of the duration of data collection procedures per data type for one participant. Time of conducting certain measurements relative to the shift hours are presented in Fig 3. Measurements related to the building layout (furniture) environmental conditions (light and sound) are recorded over the course of study when weather conditions allow. Wearable sensor data are collected continuously over different periods of time across the minimum of 7 shifts per participant. This approach aims to capture variations in staff experiences and physiological responses across different shifts and work conditions. Study includes measurements during different times of the day and different days of the week to ensure a richer data set. During this period, systematic in-person observation (observation checklist) are performed by a single researcher on different days of the week and at different times, aiming to collect a minimum of 20 hours of observation data per participant. Surveys are administered at the beginning, during and at the end of the study period assigned for each participant. Finally, debriefing includes collecting equipment, confirming data security and validity and collecting feedback from participants.

   **Surveys.**  This study uses a variety of measurement tools established by previous research and validated. Study begins with: 1) Morningness–Eveningness Questionnaire (MEQ) [46] to assess participants' chronotype and their alertness peaks and 2) The Clinical Activities Related to the Environment (CARE) scale [47]used to assess environmental affordance for nursing tasks and communication in inpatient healthcare settings. The two surveys are conducted only once, right before the first shift, for each participant. The NASA Task Load Index questionnaire [48,49] is conducted to assess the subjective mental workload of the participant at the end of each shift. Surveys aim to collect subjective data regarding staff perception of the working environment, job satisfaction, and stress levels.

   **Wearable sensor deployment.**  Three wearable sensor devices are worn by participants during working hours and outside of working hours (Fig 4). Each participant is equipped with a heart rate strap (Polar H10 Heart Rate Sensor) to collect raw heart rate data, which is later used to extract heart rate variability (HRV). The Polar H10 utilises a chest-mounted electrode system to measure R–R intervals with high temporal precision (sampling frequency: 1,000 Hz), enabling a reliable assessment of autonomic nervous system activity through time- and frequency-domain HRV

**Table 1. Duration of data collection for each data collection instrument, per participant.**

| Measurement | Data collection duration |
| --- | --- |
| Heart-rate measurement | 2-4h per shift, during behavioural mapping |
| Light dosimetry | Wrist sensor – during participant measurement period, except when exposed to water |
|  | Chest sensor – during participant measurement period, except when sleeping or exposed to water |
| Environment characteristics | Three days and one night |
| Behavioural mapping | 2-4h per shift |
| Surveys | MEQ – beginning of participant measurement period |
|  | CARE -beginning of participant measurement period |
|  | NASA Task Load Index – at the end of each shift |
| Semi-structured interviews | Once – at the end of participant measurement period |

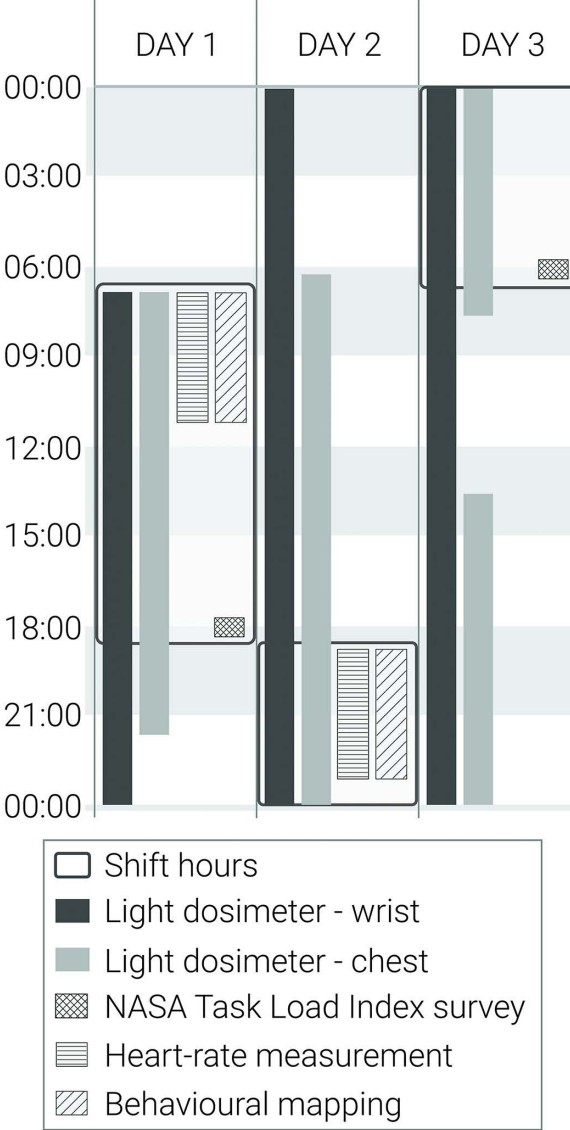

**Fig 3. Duration of data collection for each data collection instrument, over the period of three days.**

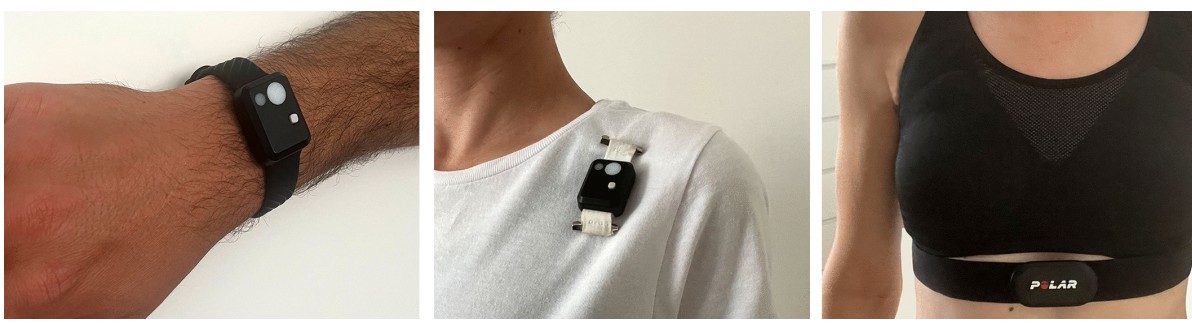

**Fig 4. Light dosimeters (left and centre), Heart rate strap (right).**

parameters. Heart rate strap is worn during systematic in-person observation hours, during the shift. Furthermore, participants wear two light dosimeters, one worn as a watch and another one as a badge on their clothes (close as possible to the eye level, but not in a way to disturb them or patients). Each light dosimeter records ambient light intensity (illuminance in lux) and melanopic equivalent daylight illuminance (melanopic EDI), providing continuous quantitative data on the spectral and circadian-effective characteristics of participants' environmental lighting across different times of day and activity settings. The light-dosimeter on the wrist is worn continuously, except when in contact with water (swimming, showering, etc), including during sleep. Participants wear chest-mounted light dosimeter all the time except when asleep or exposed to water (swimming, showering, etc). In other words, light dosimeters are worn outside of working hours to obtain the most comprehensive possible data on participants' light exposure. Two identical devices are deployed for distinct purposes. The wrist sensor, positioned directly on the body, provides more reliable activity data, which serves as a control for the observations. In contrast, the chest sensor, while more precise in recording light exposure data, yields less reliable activity measurements due to its placement on clothing rather than directly on the body. In addition to light intensity, both devices contain accelerometers that detect motion and posture, enabling differentiation between active and resting states and validating observation-based activity classifications. Additionally, this dual-sensor approach allows the study to assess which position, wrist or chest, is more comfortable for nurses to wear during their shifts. Participants are allowed to remove the devices whenever they wish. Device activity logs include "off-wrist" detection, enabling the identification of periods when the sensors are not worn and ensuring data quality control. Monitoring the battery life of the devices is facilitated by the researchers. The heart rate strap and light dosimeters are provided by the researchers, and all data monitoring and recording are performed by researcher team. Although the devices are designed for comfortable wear, participants may remove them in case of discomfort.

Prior to data collection, researchers conduct an informative session with participants to explain how the devices function. This session aims to ensure that participants are comfortable using the equipment throughout the study period. The study includes one participant at a time. Participant handling of the equipment will be minimal, as the devices used have internal memories and can store data. Furthermore, devices are rugged and designed to minimise the risk of damage in the context of this study.

Baseline measurements are needed only for HRV in order to understand the reference to which stress levels can be compared. HRV baseline is recorded on two consecutive days at the beginning of the shift, and participants are instructed to be alone and use relaxing techniques that suit them. Measurements are taken for 6 minutes, as literature shows that even shorter recordings are considered to be sufficient to obtain a valid measurement [50–52](Thong, et al, 2003; Baek, et al, 2015; Burma, et al, 2021). Light exposure is measured continuously, and no baseline is required for the purposes of this study.

**Physical environment measurement and documentation.** Output of physical environment measurements are floor plans, photographs, light data, sound data and detailed notes. Researchers use a laser measurement tool, lux meter and sound metre and conduct measurements in an unobtrusive manner, during the hours when there is the least disturbance to staff and patients and when weather conditions allow. These measurements are designed to capture objective, quantifiable environmental parameters that can be compared with participants' physiological and subjective data. Collecting repeated measures under different temporal and meteorological conditions allows for the assessment of environmental variability and supports the reliability of the dataset.

Room dimensions are taken from the provided documentation including floor plans and sections. Additional measurements are taken on-site to confirm and compare some of the dimensions and to include elements that are not present in the floorplans or that were moved or added (furniture, plants etc.). Spatial mapping of the environment provides essential contextual data for interpreting light and sound conditions within the nurses' daily workspace. Lighting conditions are mapped under specific lighting conditions and the measurements are taken via lux meter. Measurements are taken four times during the day (morning right after sunrise, 12:00, 16:00, and after sunset), under three different weather conditions

(sunny day, light overcast day, cloudy day) and one time during night hours when artificial light is on to create a baseline mapping and get a clearer understanding of the lighting conditions in the participants working environment. Commercial lux meter with internal memory is used. Sound level recordings are conducted over the period of 5 randomly selected days four times during the day (morning right after sunrise, 12:00, 16:00, and after sunset) and one time during the night. Commercial sound level metres with internal memory are used. Physical environment measurements serve to understand baseline conditions of the nurses' working environment.

**Systematic in-person observation.** Each participant is systematically observed during their working hours. Researchers randomly select hours for observation with the aim to collect a minimum of 20 hours of observation data per participant. The selected data collection method is systematic in-person observation, which enables the collection of quantitative real-time data on the behaviours and interactions of a single participant at a time. The observation data is collected by a single researcher with the use of the previously prepared data collection checklist (on paper), where observed pre-determined categories are listed and tracked. The locations of all observed activities are also mapped on the ward's floor plan. The researcher observing the participants keeps an unobtrusive role and does not interfere with the care processes in the facility. Observation findings serve to understand the movement and interaction of nurses during observed hours of their shifts and compare these findings with the HRV measurements that provide insight in stress levels. Fig 5 shows the observation sheet with the checklist used to track the participants' activities during observations. This provides a transparent account of data collection and enables replication of the method in future studies. The complete observation sheet used for the study also contains the floorplans of the psychiatric facility used to map the locations of activities; for confidentiality reasons, these cannot be published.

**Ruggedness and scalability.** Using sensors with the nursing staff of psychiatric hospitals is challenging for many reasons. They are in constant contact with patients or performing tasks that require them to be at different places, move around, not being burdened by taking care of the equipment they wear. Furthermore, patients can be aggressive, or require help that demands physical activity from nurses. Therefore wearable devices have to have long battery life, can easily be removed or be resilient to physical contact (falling off, hitting objects) and not uncomfortable to wear or disturb. Participants do not have to think about how they will store them and they have to be able to leave them in their locker, bag or desk and not think if there is a specific way of storing them. These challenges have been identified in previous studies, highlighting a gap in the ruggedness of devices to withstand potential on-site changes, partial equipment removal, and the need for quick data recovery [53,54]. Keeping in mind that it is not possible to address all challenges, this study offers a solution that is sufficiently robust, but that can at the same time provide an abundance of data. Limitations in terms of data collection exist, but they are selected as a positive trade-off with the comfort and feasibility in order to gain acceptance from participants.

## Qualitative data

**Semi-structured interviews.** Protocol includes conducting semi-structured interviews with participants to explore their experiences, perceptions, and challenges related to the physical environment of the psychiatric ward. Interviews address key themes such as spatial layout, lighting, interior, and overall comfort and designed to last approximately one hour. The questions are structured into four major topics: (1) general questions about the physical work environment, (2) patient care and communication, (3) team collaboration, communication, and job satisfaction and (4) changes to the physical environment, supports and perceived impacts. The full interview guide can be found in the supplementary material, ensuring transparency and facilitating reproducibility of the study. Interviews are recorded and transcribed verbatim. We use a RØDE Wireless GO II microphone system for all interviews to minimise ambient noise and maintain consistent audio quality with minimal intrusion on participants. In this study, we do not collect field notes or systematic interviewer observations; analyses are conducted on verbatim transcripts only.

Interviews are conducted by architects/healthcare design researchers trained in qualitative methods with no supervisory or clinical relationships to participants. To mitigate potential disciplinary bias, we use a neutral, standardised interview

**CARESCAPE - Klinik Hietzing, Psychiatrie**

Participant no. _________________          Sheet no. _________________
Gender _________________          Date _________________
Observer _________________          Time _________________

| Position | 1 | 2 | 3 | 4 | 5 | 6 | Observations/additional notes: |
|---|---|---|---|---|---|---|---|
| Time | | | | | | | |
| **commmunication** | | | | | | | |
| A  a. verbal | | | | | | | |
| b. non-verbal | | | | | | | |
| c. face-to-face | | | | | | | |
| B  d. from distance | | | | | | | |
| e. via phone | | | | | | | |
| f. other (specify) | | | | | | | |
| **activity level** | | | | | | | |
| g. sitting | | | | | | | |
| C  h. standing | | | | | | | |
| i. walking | | | | | | | |
| j. other (specify) | | | | | | | |
| **person** | | | | | | | |
| k. nurse | | | | | | | |
| l. physician | | | | | | | |
| D  m. patient | | | | | | | |
| n. family | | | | | | | |
| o. other (specify) | | | | | | | |
| **care provision** | | | | | | | |
| p. bedside | | | | | | | |
| q. in the room | | | | | | | |
| E  r. other (specify) | | | | | | | |
| s. verbal | | | | | | | |
| t. physical | | | | | | | |
| u. medication | | | | | | | |
| **location** | | | | | | | |
| G  v. inside | | | | | | | |
| w. outside | | | | | | | |
| **additional (to be added by observer)** | | | | | | | |
| x. | | | | | | | |
| H  y. | | | | | | | |
| z. | | | | | | | |

**Fig 5. Observation sheet for systematic in-person observations.**

guide and cross-check interpretations in team discussions. To further mitigate potential bias, the interviewer maintains a neutral tone and avoids leading questions, ensuring that the interviewer's assumptions or expectations do not influence participants' responses. The study continues interviews until qualitative data saturation is reached, with preliminary analysis conducted to identify when no new themes emerge, potentially increasing the sample size if necessary to ensure a comprehensive understanding of participant experiences.

## Data analysis

Collected quantitative and qualitative data is analysed separately and then triangulated to provide comprehensive answers to the research questions. The aim is to derive insights from individual datasets or through integrating these results. Triangulating findings from different data sources enhances the validity of the results; in cases where findings show divergent tendencies, new research questions may be formulated to explore these discrepancies.

To analyse heart rate variability (HRV), both time-domain and frequency-domain analyses are performed to assess autonomic nervous system function and stress levels. Time-domain metrics, such as the standard deviation of NN intervals (SDNN), help quantify overall HRV, while frequency-domain analysis explores the balance between sympathetic and parasympathetic activity. The results of the HRV analysis are statistically correlated with findings from systematic in-person observations using methods like cross-correlation analysis or Spearman's rank correlation. This helps link specific environmental conditions or participant actions with fluctuations in HRV, thereby identifying potential spatial triggers for increased or decreased stress levels.

Data from light dosimeters is analysed to calculate key metrics such as total light exposure, average light intensity, and the timing of exposure. A longitudinal model is used to analyze the relationship between light exposure patterns and stress levels derived from heart rate variability (HRV). This analysis aims to establish whether particular light conditions significantly predict stress. Sleep patterns, also obtained from the sensors, are analysed for duration and quality, with mixed-effects models used to examine potential associations with stress levels, accounting for repeated measures from individual participants.

The qualitative data from participant surveys is analysed to assess perceived workload and chronotype. Descriptive statistics is used to summarize participants' responses, while Fisher's exact tests will assess associations between perceived workload categories and chronotypes.

For the interview data, a thematic analysis is conducted on the transcripts, involving the systematic coding of data, followed by identifying, reviewing, and defining key themes. This process uses a combination of inductive and deductive approaches to identify patterns relevant to stressors and coping mechanisms. NVivo or similar qualitative analysis software may be employed to facilitate this process. Findings from qualitative interviews are then compared to quantitative data to enhance the overall interpretation of results. For example, themes emerging from the interviews regarding perceived stressors are compared with light exposure and HRV metrics to identify convergent or divergent insights.

The discussion interprets these findings in light of existing literature on healthcare environments and their impact on staff well-being, exploring future research and practice implications. Areas of the agreement should provide more robust evidence, while divergent findings will highlight gaps or inconsistencies that will define future research directions.

To address missing data in the study, various strategies are employed to ensure the integrity and completeness of the analysis. First, data quality is monitored throughout the collection process, and any missing or incomplete quantitative data (e.g., from heart rate variability measurements or light dosimeters) is evaluated to determine whether the data is missing at random or systematically. For missing quantitative data, the multiple imputation method is applied to account for gaps without introducing bias into the results. In cases where large portions of data are missing from certain participants, sensitivity analyses are conducted to evaluate the impact of the missing data on the overall findings.

For qualitative data, if interview or survey responses are incomplete, thematic analysis proceeds with the available data, while noting any areas where missing responses might limit interpretative depth. Any discrepancies between the qualitative and quantitative datasets due to missing data are carefully considered during the triangulation process, ensuring that missing elements did not skew the overall interpretation of the findings.

## Data management

This study is committed to ensuring both secure data management and the promotion of open science principles. All data collected throughout the study are initially stored on secure acquisition devices or systems provided by the institutions

involved (TU Wien and 2nd Department of Psychiatry and Psychotherapy, Hietzing Clinic, Vienna Healthcare Group), ensuring that the data is managed in a secure manner and protected from unauthorized access or loss. Strict security measures, such as encryption and backups, are implemented to maintain data integrity throughout the study's duration. Access to the stored data is tightly controlled and restricted solely to authorized researchers directly involved in the study. This ensures confidentiality, with access granted on a need-to-know basis, and only those with the necessary credentials are allowed to manipulate the data. In addition to securing the data, the study aligns with the principles of open science, aiming to enhance transparency and accessibility in research. Once the data is anonymized and deemed publication-ready, it will be made available in open data repositories, enabling other researchers to verify results, conduct further analyses, and build upon the findings. The anonymized data intended for publication will be uploaded to open data repositories once all necessary preparations are complete. While the data will generally be made available for public access, some restrictions may apply based on ethical considerations, data sensitivity, or researcher preferences, in which case access may be limited to a specific group of individuals. This approach ensures both secure handling of sensitive data and supports the broader scientific community by fostering collaboration and promoting research transparency.

## Patient and public involvement

Patients and the public were not involved in this study.

## Ethical considerations and bias control

The study complies with all ethical regulations required for responsible and ethical research. TU Wien provides services for researchers who involve low-risk human participation; these include a personal research ethics consultation by the Unit of Responsible Research Practices and a consultative ethics committee. At TU Wien submitting studies for the ethics committee's review is voluntary and the peer review process does not result in an approval. The first author of this paper made use of the personal research ethics consultation, where the potential risks and the benefits for the study participation were identified and addressed and the informed consent documentation was discussed in detail. The recommendations of this consultation were implemented in the study, its documentation and reporting. A rigorous protocol is implemented to ensure comprehensive data anonymization. Each participant is assigned a randomly generated unique identifier at the study's outset, after which all personally identifiable information (PII) is permanently removed from the dataset. The critical link between unique identifiers and participants' personal details is immediately and irreversibly destroyed following data collection. Furthermore, data is systematically aggregated and generalized to prevent any potential indirect identification. A thorough risk assessment is conducted to identify and mitigate any possible re-identification pathways through complex data combinations. All research team members in contact with data underwent specialized training in data protection. These multilayered measures guarantee that the research data remains completely anonymous, rendering individual participant re-identification impossible. Participants are allowed to drop out at any point. Furthermore each participant is provided with informed consent detailing the study's purpose, procedures, and potential risks. Researchers ensure that each participant gets an in-person session where all aspects of the study are explained. The ethical documents outlining the study and consent form explicitly mention the possibility of participants interacting with patients, as they may indirectly contribute to the observations. To eliminate any risk of observational bias while upholding the highest standards of patient confidentiality and respect, the ethical documents explicitly state that participants may interact with patients during the study. However, strict measures are in place to ensure that patient identities cannot, under any circumstances, be identified. These measures include removing all personally identifiable information, ensuring that no data collected during interactions can be traced back to individual patients, and implementing protocols to prevent indirect identification through data combinations. Researchers in the study who are in contact with the data received comprehensive training on patient confidentiality and ethical conduct, including handling sensitive information appropriately. Additionally, all researcher's actions during the study (i.e., observations) are designed to minimize any impact on

patients, ensuring they are not disturbed or inconvenienced during observations. Oversight mechanisms are in place to monitor compliance with these protocols throughout the study. These robust precautions ensure that patient privacy is fully protected and their well-being is prioritized at all times. No personal data will be collected during in-person observations, and observations are recorded using pen and paper without any form of video or audio recording. Only general activity types, interaction counts, and locations will be recorded using predefined categories to ensure consistency and reduce researcher bias. To mitigate response bias, participants are encouraged to share their experiences freely, and they can request the omission or deletion of data if they feel uncomfortable during sensitive situations. Participants are granted full transparency and can access the collected data at any point to ensure they agree with its content. To control for potential bias in data interpretation, both quantitative and qualitative data will be analyzed independently by multiple researchers, and discrepancies in findings will be discussed to reach a consensus. Finally, researchers from the psychiatric hospital will review all collected data in anonymized form before it is used in the analysis, ensuring that data integrity is maintained while protecting participants' privacy and the rights of all involved. This review process further safeguards against any potential biases that might arise from data misinterpretation or ethical concerns.

## Discussion and conclusion

The study protocol outlined in this paper details a convergent parallel mixed-method design aimed at addressing research gaps concerning the physical environment's impact on psychiatric nurses Well-Being and Professional Interactions. The primary goal is to provide a comprehensive methodology investigating the relationship between environmental factors and nurses' well-being, emphasising stress, interactions, and overall perception of the work environment. Importantly, the research methodology is designed to be reproducible, either in full or partially, to facilitate the collection of data critical for understanding environmental impacts in psychiatric settings.

The application of sensors in psychiatric hospitals is challenging due to the need for robustness, including physical durability, battery life, and user comfort, which narrows the range of suitable devices. Additionally, contextual interpretation of non-verbal and environmental cues during interviews is limited as we did not collect interviewer observation notes. Furthermore, although the study's small sample size may raise concerns about broad statistical generalizability, it is justifiable within mixed-method research that emphasises the depth and richness of data. Finally, the presence of researchers during the systematic in-person observations may have impacted participants' behaviour, a phenomenon known as the Hawthorne effect. These limitations have been thoroughly considered and addressed.

Although this paper primarily outlines the study design, the anticipated outcomes include identifying environmental characteristics, such as lighting, acoustics, and spatial layout, that affect nurses' stress, interactions, and well-being. Integrating qualitative and quantitative data will provide insight into how these factors interact in psychiatric settings. Based on these insights, the study aims to develop evidence-based recommendations to improve psychiatric facility design, focusing on reducing occupational stress, enhancing staff collaboration, and supporting therapeutic quality. The proposed methodological framework may also serve as a model for similar research in other healthcare contexts.

Furthermore, the findings from this study have the potential to reveal new relationships between factors influencing stress levels, interactions, and nurses' perceptions of their work environment in psychiatric hospitals. Such insights serve as a foundation for evidence-based recommendations to enhance the design of psychiatric facilities, ultimately improving the well-being of nursing staff and patients.

The findings from the study will be shared with all interested audiences. Additionally, they will validate the methodology and provide a foundation for future research.

## Supporting information

**S1 File. Reporting Guidelines Checklists (GRAMMS, STROBE, COREQ).**
(DOCX)

**S2 File. Semi-structured interview guide.**
(DOCX)

## Acknowledgments

We would like to extend our gratitude to Anastasia Dombrovska for her invaluable support during the recruitment process.

## Author contributions

**Conceptualization:** Milica Vujovic, Maja Kevdzija, Friedrich Neuhauser, Matthäus Fellinger.

**Data curation:** Milica Vujovic, Matthäus Fellinger.

**Funding acquisition:** Milica Vujovic.

**Investigation:** Milica Vujovic, Maja Kevdzija, Matthäus Fellinger.

**Methodology:** Milica Vujovic, Maja Kevdzija, Friedrich Neuhauser, Matthäus Fellinger.

**Project administration:** Milica Vujovic, Matthäus Fellinger.

**Resources:** Milica Vujovic.

**Supervision:** Milica Vujovic, Maja Kevdzija, Matthäus Fellinger.

**Validation:** Milica Vujovic, Maja Kevdzija, Friedrich Neuhauser, Matthäus Fellinger.

**Writing – original draft:** Milica Vujovic, Maja Kevdzija, Friedrich Neuhauser, Matthäus Fellinger.

**Writing – review & editing:** Milica Vujovic, Maja Kevdzija, Friedrich Neuhauser, Matthäus Fellinger.

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
