## [Decision Letter · Decision Letter 0]

14 Oct 2025

Dear Dr. Vujovic,

Thank you for submitting your manuscript to PLOS ONE. After careful consideration, we feel that it has merit but does not fully meet PLOS ONE’s publication criteria as it currently stands. Therefore, we invite you to submit a revised version of the manuscript that addresses the points raised during the review process.

We look forward to receiving your revised manuscript.

Kind regards,

Javier Fagundo-Rivera, PhD

Academic Editor

PLOS ONE

Journal Requirements:

“TU Wien Bibliothek provides financial support through its Open Access Funding Programme.”

4. In the online submission form, you indicated that “No datasets were generated or analysed during the current study. All relevant data from this study will be made available upon study completion.”

6. Please ensure that you include a title page within your main document. You should list all authors and all affiliations as per our author instructions and clearly indicate the corresponding author.

**Additional Editor Comments** :

Dear Authors,

Thank you for submitting your manuscript to PLOS ONE. Your paper has now been reviewed by independent experts. After considering their reports and my own evaluation, I am writing to inform you that the editorial decision is Major Revision.

The reviewers agree that this is an original and well-designed study protocol addressing an important topic: the influence of the physical environment on psychiatric nurses’ well-being and professional interactions. The manuscript is clearly structured, written in intelligible English, and compliant with accepted ethical and reporting standards.

However, several aspects require improvement before the paper can be considered for publication. Specifically, the methods section should be refined to improve sentence structure and clarity, terminology should be used consistently throughout, and minor grammatical issues should be corrected.

We therefore invite you to revise your manuscript thoroughly in light of these comments. Please include a detailed response to the reviewers’ points and submit a revised version highlighting the changes made.

We look forward to receiving your revision and appreciate your continued commitment to strengthening this valuable contribution.

With best regards.

Reviewer's Responses to Questions

**Comments to the Author**

1. Does the manuscript provide a valid rationale for the proposed study, with clearly identified and justified research questions?

Reviewer #1: Yes

Reviewer #2: Yes

Reviewer #3: Partly

2. Is the protocol technically sound and planned in a manner that will lead to a meaningful outcome and allow testing the stated hypotheses?

Reviewer #1: Yes

Reviewer #2: Partly

Reviewer #3: Partly

3. Is the methodology feasible and described in sufficient detail to allow the work to be replicable?

Reviewer #1: No

Reviewer #2: No

Reviewer #3: No

4. Have the authors described where all data underlying the findings will be made available when the study is complete?

Reviewer #1: Yes

Reviewer #2: No

Reviewer #3: No

5. Is the manuscript presented in an intelligible fashion and written in standard English?

Reviewer #1: Yes

Reviewer #2: Yes

Reviewer #3: Yes

**Reviewer #1: SEE DOCUMENT ATTACHED**

The manuscript presents an original study protocol investigating how the physical environment affects psychiatric nurses’ well-being and professional interactions. The manuscript is clearly structured, written in intelligible English, and adheres to accepted ethical and reporting standards.

Suggested improvements include refining sentence structure in the methods section, ensuring consistent terminology, and correcting occasional grammatical issues. Once these major revisions are made, the article will be suitable for publication.

**Reviewer #2: **

The study protocol titled “Investigating the Influence of the Physical Environment on Psychiatric Nurses Well-Being and Professional Interactions: A Convergent Parallel Mixed-Method Study Protocol” is well-written and informative. Below are some comments (mostly statistical) to the authors:

1. Line 154: Can you please elaborate on what sense the methodology is reproducible? If the methodology is using convenience sampling, can this be reproducible? Even if it was using any statistical sampling technique, the same study participants may not be repeatedly selected for reproducibility purposes.

2. Line 169: Four different units out of how many total units? What are the reasons for choosing only four units and why/how they were chosen? Please clarify.

3. Line 170: any specific reason for writing these age groups separately, instead of "age 18 and above"?

4. Line 172: By "...all units", did you mean the four selected units? Please, paraphrase for clarity if so.

5. Line 200: In the supplement, the authors mentioned “In the Methods and Analysis section, we justify the target sample size of 20 participants based on comprehensive data collection methods, including in-depth interviews.” However, this justification falls short of a rigorous statistical approach. Please elaborate on how the sample size is justifiable in the absence of any statistical hypothesis, related power analysis, and potential participant dropout considerations.

**Reviewer #3: **

This is a very interesting write up. The abstract was very enlightening you did a good job on it.

1.In the Study Setting, can you be more detailed in the explanation of the study setting

2.In Study Participants line 218 and 219 don't seem together or continuous kindly check.

3. Can you explain what you mean in Line 243 and possibly rephrase?

4. Kindly replace insure with ensure

5.With a writeup of this quality there needs to me more tidying with the result and if possible, recommendations which was not included. The article would be more qualitative with those included. Thank you

**Do you want your identity to be public for this peer review?** For information about this choice, including consent withdrawal, please see our Privacy Policy

Reviewer #1: No

Reviewer #2: No

Reviewer #3: **Yes: ** Folashade Onatola Toye

---

## [Author Response · Author response to Decision Letter 1]

21 Nov 2025

Dear Reviewers,

Thank you very much for taking the time to review our paper. We have carefully considered all of your comments and revised the manuscript accordingly. Your suggestions have been extremely helpful in improving the quality of our work. All detailed responses can be found in the accompanying “Response to Reviewers” document.

Best regards,

Authors

---

## [Decision Letter · Decision Letter 1]

21 Dec 2025

Investigating the influence of the physical environment on psychiatric nurses wellbeing and professional interactions: a convergent parallel mixed-method study protocol

PONE-D-25-06162R1

Dear Dr. Vujovic,

We’re pleased to inform you that your manuscript has been judged scientifically suitable for publication and will be formally accepted for publication once it meets all outstanding technical requirements.

Kind regards,

Javier Fagundo-Rivera, PhD

Academic Editor

PLOS One

Additional Editor Comments (optional):

Dear Authors,

You have satisfactorily responded to all comments and the manuscript has been improved.

I recommend to Accept this article.

Reviewers' comments:

Reviewer's Responses to Questions

**Comments to the Author**

1. Does the manuscript provide a valid rationale for the proposed study, with clearly identified and justified research questions?

Reviewer #1: Yes

2. Is the protocol technically sound and planned in a manner that will lead to a meaningful outcome and allow testing the stated hypotheses?

Reviewer #1: Yes

3. Is the methodology feasible and described in sufficient detail to allow the work to be replicable?

Reviewer #1: Yes

4. Have the authors described where all data underlying the findings will be made available when the study is complete?

Reviewer #1: Yes

5. Is the manuscript presented in an intelligible fashion and written in standard English?

Reviewer #1: Yes

You may also provide optional suggestions and comments to authors that they might find helpful in planning their study.

Reviewer #1: The revised manuscript presents a well-structured and methodologically robust study protocol. The authors have made substantial improvements in clarity, methodological transparency, and integration of mixed-methods procedures. The manuscript now adheres closely to reporting standards and demonstrates clear potential to contribute meaningfully to interdisciplinary research in mental-healthcare design.

**Do you want your identity to be public for this peer review?** For information about this choice, including consent withdrawal, please see our Privacy Policy

Reviewer #1: No

---

## [Editor Report · Acceptance letter]

PONE-D-25-06162R1

PLOS One

Dear Dr. Vujovic,

I'm pleased to inform you that your manuscript has been deemed suitable for publication in PLOS One. Congratulations! Your manuscript is now being handed over to our production team.

Kind regards,

on behalf of

Dr. Javier Fagundo-Rivera

Academic Editor

PLOS One